# AttendLight: Universal Attention-Based Reinforcement Learning Model for Traffic Signal Control

**Afshin Oroojlooy**
SAS Institute Inc.
Cary, NC 27513
afshin.oroojlooy@sas.com

**Mohammadreza Nazari**
SAS Institute Inc.
Cary, NC 27513
reza.nazari@sas.com

**Davood Hajinezhad**
SAS Institute Inc.
Cary, NC 27513
davood.hajinezhad@sas.com

**Jorge Silva**
SAS Institute Inc.
Cary, NC 27513
jorge.silva@sas.com

## Abstract

We propose AttendLight, an end-to-end Reinforcement Learning (RL) algorithm for the problem of traffic signal control. Previous approaches for this problem have the shortcoming that they require training for each new intersection with a different structure or traffic flow distribution. AttendLight solves this issue by training a single, universal model for intersections with any number of roads, lanes, phases (possible signals), and traffic flow. To this end, we propose a deep RL model which incorporates two attention models. The first attention model is introduced to handle different numbers of roads-lanes; and the second attention model is intended for enabling decision-making with any number of phases in an intersection. As a result, our proposed model works for any intersection configuration, as long as a similar configuration is represented in the training set. Experiments were conducted with both synthetic and real-world standard benchmark data-sets. Our numerical experiment covers intersections with three or four approaching roads; one-directional/bi-directional roads with one, two, and three lanes; different number of phases; and different traffic flows. We consider two regimes: (i) single-environment training, single-deployment, and (ii) multi-environment training, multi-deployment. AttendLight outperforms both classical and other RL-based approaches on all cases in both regimes.

## 1 Introduction

With the emergence of urbanization and the increase in household car ownership, traffic congestion has been one of the major challenges in many highly-populated cities. Traffic congestion can be mitigated by road expansion/correction, sophisticated road allowance rules, or improved traffic signal controlling. Although either of these solutions could decrease the travel times and fuel costs, optimizing the traffic signals is more convenient due to the limited funding resources and opportunity of finding more effective strategies. This paper introduces a framework for learning a general traffic control policy that can be deployed in an intersection of interest and ease the traffic flow.

Approaches for controlling signalized intersections could be categorized into two main classes, namely conventional methods and adaptive methods. In the former, customarily rule-based fixed cycles and phase times are determined a priori and offline based on historical measurements as well

as some assumptions about the underlying problem structure. However, since traffic behavior is dynamically changing, that makes most conventional methods highly inefficient. In adaptive methods, decisions are made based on the current state of the intersection. Self-organizing Traffic Light Control (SOTL) [12] and *Max-pressure* [28] are among the most popular adaptive methods that consider the number of approaching vehicles to the intersection in their traffic control algorithm (See, e.g., [14] for more details). These methods bring considerable improvements in traffic signal control; nonetheless, they are short-sighted and do not consider the long-term effects of the decisions on the traffic. Besides, these methods do not use the feedback from previous actions toward making more efficient decisions. In response, more sophisticated algorithms have been proposed. Using artificial intelligence (AI) for controlling traffic signals has recently attracted a lot of attention, due to the major potential benefits that it can bring toward having less-congested cities. Reinforcement Learning (RL) [26], which has been flourished in recent years in the AI community, has shown superior performance for a wide range of problems such as games [25], robotics [13], finance [11], and operations research [7], only to name a few. This coincides with growing applications of RL in traffic signal control problem (TSCP) [15, 16, 22, 29, 37].

In spite of immense improvements achieved by RL methods for a broad domain of intersections, the main limitation for the majority of the methods is that the proposed model needs to be re-designed and re-trained from scratch whenever it faces a different intersection either with different topology or traffic distribution. Learning specialized policies for each individual intersection can be problematic, as not only do RL agents have to store a distinct policy for each intersection but in practice data collection resources and preparation impose costs. These costs include the burden on human-experts' time to setup a new model, and computational resources to train and tune a new model. Thus, it is not clear whether such a cumbersome procedure is feasible for a city with thousands of distinct intersections. There exists some prior work on partially alleviating such issues [30, 37] using transfer learning; however, the trained models still need to be manipulated for different intersection structures and require retraining to achieve reasonable performance.

To address these issues, we bring ideas from attentional models [5] into TSCP. Rather than specializing on a single intersection, our goal is to design a mechanism with satisfactory performance across a group of intersections. Attentional mechanisms are a natural choice, since they allow unified system representations by handling variable-length inputs. We propose the AttendLight framework, a reinforcement learning algorithm, to train a "universal" model which can be used for any intersection, with any number of roads, lanes, phase, traffic distribution, and type of sensory data to measure the traffic. In other words, once the model is trained under a comprehensive set of phases, roads, lanes, and traffic distribution, our trained model can be used for new unseen intersection, and it provides reasonable performance as long as the intersection configuration follows a pattern present in the training set. We find that AttendLight architecture can extract an abstract representation of the intersection status, without any extra grounding or redefinition, and reuse this information for fast deployment. We show that our approach substantially outperforms purely conventional controls and FRAP [37], one of the state-of-the-art RL-based methods.

## 2   Related Work

The selection of RL components in traffic light control is quite challenging. The most common action set for the traffic problem is the set of all possible phases. In [22] an image-like representation is used as the state and a combination of vehicle delay and waiting time is considered as the reward. A deep Q-Network algorithm was proposed in [15], where the queue length of the last four samples is defined as the state, and reward is defined as the absolute value of the difference between queue length of approaching lanes. In [16] the intersection was divided into multiple chunks building a matrix such that each chunk contains a binary indicator for existence of a car and its speed. Using this matrix as the state and defining reward as the difference of the cumulative waiting time for two cycles, they proposed to learn the duration of each phase in a fixed cycle by a Double Dueling DQN algorithm with a prioritized experience replay. Likewise, [24] defined a similar state by dividing each lane into a few chucks and the reward is the reduction of cumulative delay in the intersection. A DQN approach was proposed to train a policy to choose the next phase. Ault et al. [4] proposed three DQN-based algorithms to obtain an interpretable policy. A simple function approximator with 256 variables was used and is showed that it obtains slightly worse result compared to the DQN algorithm with an uninterpretable approximator. The IntelliLight algorithm was proposed in [34]. The state

and reward are a combination of several components. A multi-intersection problem was considered in [31], where an RL agent was trained for every individual intersection. The main idea is to use the pressure as the reward function, thus the algorithm is called PressLight. To efficiently model the influence of the neighbor intersections, a graph attentional network [29] is utilized in CoLight [32]. See [33] for a detailed review of conventional and RL-based methods for TSCP.

There have been attempts to transfer the learned experiences between different intersections. For example, in FRAP algorithm [37] a trained model for a 4-way intersection needs to be changed through reducing the neurons as well as zero-padding modifications to make it compatible for a 3-way intersection. A modification of FRAP was proposed on [35]. This new algorithm is called MetaLight, where the key idea is to use meta-learning strategy proposed in [10] to make a more universal model. However, MetaLight still needs to re-train its model parameter for any new intersection. In [30], a multi-agent RL algorithm is proposed to control the traffic signals for multiple intersections with arterial streets. The key idea is to train a single agent for an individual intersection and then apply *Transfer Learning* to obtain another model for a 2-intersection structure. Similarly, a model for a 3-intersection structure can be obtained from 2-intersection and etc. Besides the natural drawbacks of Transfer Learning, such as the assumption on the data distribution, the proposed model in this work is not robust in terms of the number of approaching roads, lanes, and phases. For example, a trained model for a single 4-way intersection cannot be transferred to a 3-way one.

## 3 Traffic Signal Control Problem

We consider a single-intersection traffic signal control problem (TSCP). An *intersection* is defined as a junction of a few roads, where it can be in the form of 3-way, 4-way or it can have a more complex structure with five or more approaching roads. Each road might have one or two direction(s) and each direction includes one lane or more. Let $\mathcal{M}$ be a set of intersections, where each intersection $m \in \mathcal{M}$ is associated with a known intersection topology and traffic-data. Throughout this paper, we simplify notations by omitting their dependency on $m$.

Let us define $s_l^t$ as the *traffic characteristics* of lane $l \in \mathcal{L}$ at time $t$ , where we use $\mathcal{L}$ to denote the set of all approaching (i.e., either entering or leaving) lanes to the intersection. We denote by $\mathcal{L}^{in}$ and $\mathcal{L}^{out}$ the set of entering and leaving lanes, respectively. Clearly, $\mathcal{L}^{in} \cup \mathcal{L}^{out} = \mathcal{L}$ holds. In TSCP, different traffic characteristics for each lane are proposed in the literature, e.g., queue length, waiting time, delay, the number of moving vehicles, etc. See [33] for more details. The model we propose in this work is not restricted to any of the mentioned characteristics and assigning $s_l^t$ to any of these is completely up to the practitioner. There is a finite number of ways that the cars can transition from the entering lanes to leaving lanes. We define a *traffic movement* $v_l$ as a set that maps the entering traffic of lane $l \in \mathcal{L}^{in}$ to possible leaving lanes $l' \in \mathcal{L}^{out}$.

The set of the traffic movements which are valid during a green light is called a *phase*, and it represented by $p$ and we denote by $\mathcal{P}$ the set of all phases. An intersection has at least two phases, and they may have some shared traffic movement(s). We define the *participating lanes* $\mathcal{L}_p$ as the set of lanes that have appeared in at least one traffic movement of phase $p$. Each phase runs for a given minimum amount of time and after that, a decision about the next phase should be taken.

To further clarify our problem definition and the terminologies, we use a 3-way intersection example, depicted in Figure 1. There are six lanes entering the intersection and six leaving lanes. We have labeled these lanes with $l_k^{in}$ and $l_k^{out}$, $k \in \{1, \cdots, 6\}$ in Figure 1a. Accordingly, there are six traffic movements denoted by $v_1, \cdots, v_6$ as illustrated in Figure 1b. Also, three possible phases for this intersection along with the valid traffic movements of each phase are listed in Figure 1. For example, phase-1 involves two traffic movements $v_6 = \left\{ l_6^{in} \to l_1^{out}, l_6^{in} \to l_2^{out} \right\}$ and $v_5 = \left\{ l_5^{in} \to l_3^{out}, l_5^{in} \to l_4^{out} \right\}$, hence the set of participating lanes associated with this phase is $\mathcal{L}_1 = \{ l_5^{in}, l_6^{in}, l_1^{out}, l_2^{out}, l_3^{out}, l_4^{out} \}$. As one may notice, the number of traffic movements as well as the number of participating lanes are not necessarily the same for different phases. For example, in Figure 1 phase-1 and phase-3 involve two traffic movements, while there are three traffic movements in phase-2. Further, phase-1 and phase-3 involve six participating lanes while phase-2 includes nine lanes. This results in different size of the input/output of the model among different intersection instances. Therefore, building a universal model which handles such complexity is not straightforward using conventional deep RL algorithms. To address this issue, we design AttendLight which uses a special attention mechanism as described in the next section.

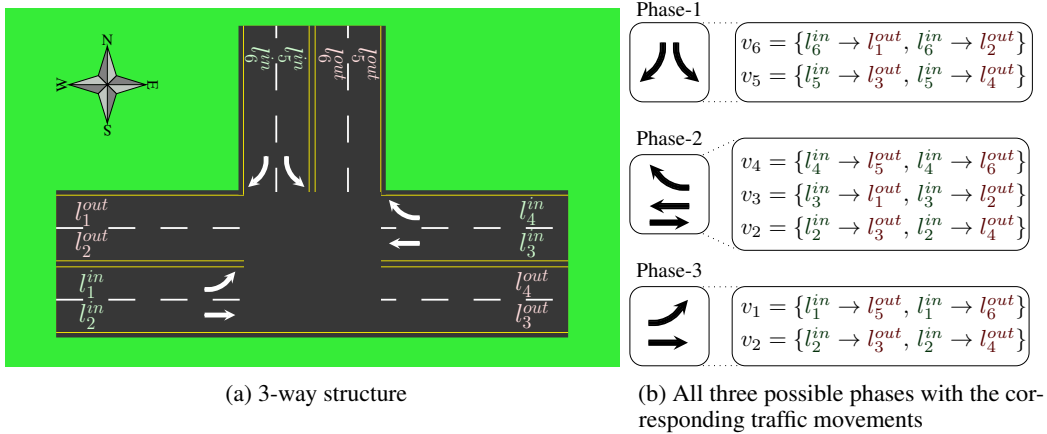

(a) 3-way structure

(b) All three possible phases with the corresponding traffic movements

Figure 1: A 3-way intersection topology with its available phases and traffic movements

The main goal of TSCP is to minimize the average travel time of all vehicles within a finite time frame. However, the travel time is not a direct function of state and action so in practice it is not possible to optimize it directly. Zheng et al. [31] use an alternative reward measure, namely *pressure*, which is defined as the absolute value of the total number of waiting vehicles in entering lanes minus the total number of leaving vehicles. They show that minimizing pressure is equivalent to minimizing the average travel time. Similarly, in this work we also choose to minimize the pressure of the intersection, though we report the average travel times in the numerical experiments.

## 4    Reinforcement Learning Design

In this section we present our end-to-end RL framework for solving the TSCP. To formulate this problem into an RL context, we first require to identify *state*, *action*, and *reward*.

**State**. The state at time $t$ is the traffic characteristics $s_l^t$ for all lanes $l \in \mathcal{L}$, i.e., $s^t = \{s_l^t, l \in \mathcal{L}\}$.

**Action**. At each time step $t$, we define the action as the active phase at time $t + 1$.

**Reward**. Following the discussion in [31], the reward in each time step is set to be the negative of intersection pressure.

In TSCP, we are interested in learning a policy $\pi$, which for a given state $s^t$ of an intersection suggests the phase for the next time-step in order to optimize the long-term cumulative rewards. We design a unified model for approximating $\pi$ that fits to every intersection configuration. The AttendLight model that we present in Section 4.2 instantiates such a policy $\pi$ that achieves this universality by appropriate use of two attention mechanisms.

### 4.1    Attention Mechanism

Attention mechanism introduced for natural language processing [5, 17], but it has been proved to be effective in other domains such as healthcare [8], combinatorial optimization [6, 19], and recommender systems [23]. By providing a mechanism to learn the importance of each element in a problem with variable number of inputs, attention allows having a fixed length encoding of inputs. The attention mechanism that we use in this work is the key for achieving the universal capability of the AttendLight. We adjust the attention mechanism of [5] for solving TSCP. In this setting, we have two types of inputs to $\texttt{attention}(\cdot, \cdot)$, namely a set of references $\{r_i\}$ and a query $q$. The attention computes an *alignment* $a \coloneqq \{a_i\}$, where

$$a_i = u_a \texttt{tanh}(U_r \bar{r}_i + U_q \bar{q}), \tag{1}$$

in which $\bar{q}$ and $\bar{r}_i$ are trainable linear mappings of $q$ and $r_i$, respectively; $U_q, U_r, u_a$ are trainable variables. Note that $a$ has the same size as the input reference set $\{r_i\}$. Finally, the attention returns $w$, which is a probability distribution computed as $\texttt{softmax}(a)$.

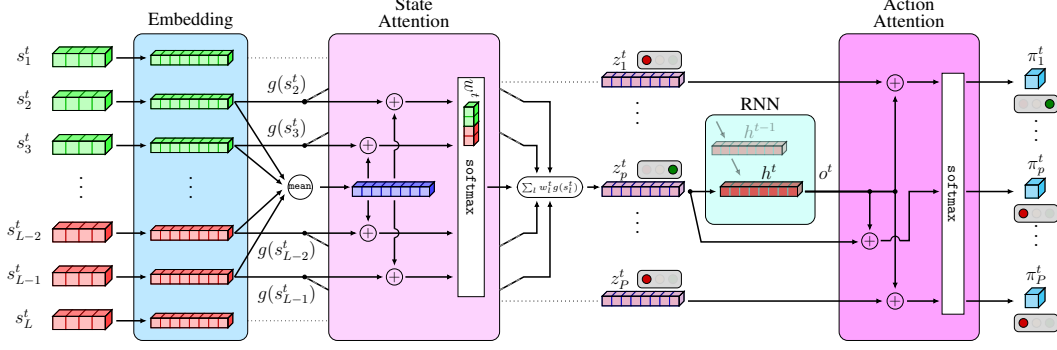

Figure 2: AttendLight model

## 4.2 AttendLight Algorithm

AttendLight is our proposed algorithm and has two major responsibilities: *i)* extracting meaningful *phase representations* $z_p^t$ for every phase $p$, and *ii)* deciding on the next active phase. To add universality to these responsibilities, the input and output dimension of the model needs to be independent of intersection configuration. To this order, we propose two attention mechanisms—as introduced in Section 4.1—called `state-attention` and `action-attention` for handling the phase representation from the raw-state and for choosing the next phase, respectively. The policy model that we used in AttendLight is visualized in Figure 2. Next, we explain how AttendLight achieves these goals.

A key part of AttendLight is to construct meaningful phase representations. Although identifying the phase representations for TSCP is not a trivial process, we expect that it could be extracted from the traffic characteristic of the participating lanes $\{s_l^t, \, l \in \mathcal{L}_p\}$. For example, the number of cars in participating lanes can be utilized to infer how congested the traffic movements are, how close the cars are with respect to the intersection, and whether or not activating a phase will ease the traffic. AttendLight uses an embedding followed by an attention mechanism to learn the phase representation.

In order to allow more capacity in feature extraction, we embed $s_l^t$ into a higher dimensional space. In this work, we use a single `Conv1D` transformation as our embedding function $g$, but one may use more intricate embedding functions as well. The `state-attention` uses these embeddings and returns the importance weights $w_p^t := \{w_l^t, \, l \in \mathcal{L}_p\}$ for attending on $g(s_l^t)$ of each participating lane in phase $p$, i.e.,

$$w_p^t = \texttt{state-attention}\left(r_p^t, q_p^t\right), \, \forall p \in \mathcal{P}, \tag{2}$$

where the references $r_p^t$ are the embedded participating lane characteristics and the query $q_p^t$ is their average, i.e.,

$$r_p^t := \left\{g(s_l^t), \, l \in \mathcal{L}_p\right\} \qquad q_p^t := \sum_{l \in \mathcal{L}_p} \frac{g(s_l^t)}{|\mathcal{L}_p|}.$$

Then, the phase representation $z_p^t$ is computed as $z_p^t = \sum_{l \in \mathcal{L}_p} w_l^t \times g(s_l^t)$. From this formulation, we can see that the phase representation is independent of the number of participating lanes in each phase since $\mathcal{L}_p$ can have any arbitrary cardinality. It worth mentioning that using the average of $g(s_l^t)$ as the query $q_p^t$ in the state-attention, allows the model to learn the importance of each lane-traffic compared to the average traffic. For example, whenever all lanes are either full or mostly empty, the query detects the situation and in turn, the attention model assigns appropriate weights to each lane.

To capture the sequential information of active phases, we incorporate an LSTM cell. At every time step, the phase representation $z_p^t$ of the active phases, i.e., the phase associated with the current green light is fed into the cell. LSTM uses a hidden memory $h^t$ to store encoded sequence of active phase in previous times and produces an output $o^t$. Then, the `action-attention` model provides the probability $\pi^t := \{\pi_p^t, \, p \in \mathcal{P}\}$ of switching to any of the phases in the next time step using

$$\pi^t = \texttt{action-attention}\left(\{z_p^t, \, p \in \mathcal{P}\}, o^t\right), \tag{3}$$

Hence, AttendLight provides an appropriate mapping of the states to probability of taking actions for any intersection configuration, regardless of the number of roads, lanes, traffic movements, and

number of phases. We would like to emphasize that AttendLight is invariant to the order of lanes or phases, so how to enumerate these components will result in the same control decisions.

## 4.3 RL Training

We train the AttendLight with a variance-reduce variant of REINFORCE [27]. This training algorithm is quite standard, so we leave its detailed description to Appendix A.3. To train the AttendLight, we follow two regimes for the intersection sampling process: (i) train for a single environment and deploy on the same environment, which we refer to it as "*single-env*" regime, and (ii) train on multiple environments and deploy on multiple environments, which we call is as "*multi-env*" regime. In the first regime, we use a particular environment instance $m$ to train a policy and deploy the model on the same environment instance $m$. This is the common practice in all of the current RL algorithms for TSCP [31, 35, 37], in which the trained model only works on the intended problem. While in the second regime, in each episode we sample $n$ environments from $\mathcal{M}$ and run the train-step based on all those environments. In experiments of this paper, we let $n = |\mathcal{M}|$. However, in Appendix B, we present alternative multi-env regimes to deal with a large number of intersections.

## 5 Numerical Experiments

Many variants of the intersections have been studied in the literature and we claim that a single model can provide reasonable phase decisions for all of them. We consider 11 intersection topologies, where they vary in terms of the number of approaching roads (i.e., 3-way or 4-way), and the number of lanes in each road. Further, each of these 11 intersections may have a different number of phases and traffic-data. Table 1 summarizes the properties of all intersections.

To train and test AttendLight, a combination of real-world and synthetic traffic-data is utilized. For 4-way intersections with two lanes, we use the real-world traffic-data of intersections in Hangzhou and Atlanta [33, 37]. For notation simplicity, we denote these data by H1, $\cdots$, H5 and A1, $\cdots$, A5 for Hangzhou and Atlanta, respectively. For the rest of the intersections with two lanes (e.g., 3-way intersections), slightly adapted version of these data-sets are used. Due to lack of real-world data for the intersection with three lanes, we created synthetic traffic-data denoted by S1, $\cdots$, S6. The combination of intersection topologies, their available phases, and traffic-data allows us to construct the set $\mathcal{M}$ with 112 unique intersection instances. We label each intersection instance by INT#-dataID-#phase. For example, the problem which runs intersection INT5 with the H1 traffic-data through 8-phase is denoted by INT5-H1-8. Table 1 summarizes the properties of all 11 intersections used in this study.

Table 1: All intersection configurations

| intersection ID | INT1 | INT2 | INT3 | INT4 | INT5 | INT6 | INT7 | INT8 | INT9 | INT10 | INT11 |
|---|---|---|---|---|---|---|---|---|---|---|---|
| #road | 3 | 3 | 3 | 3 | 3 | 3 | 4 | 4 | 4 | 4 | 4 |
| #lanes per road | 2 | 2 | 1,2 | 1,3 | 1,3 | 1,3 | 2 | 2 | 3 | 2,3 | 2,3 |
| #phase | 3 | 3 | 2 | 3 | 3 | 3 | 4,8 | 3 | 4,8 | 4,8 | 3,5 |
| $(\min_p |\mathcal{L}_p|, \max_p |\mathcal{L}_p|)$ | (1,2) | (1,2) | (1,2) | (2,4) | (2,4) | (2,4) | (2,2) | (2,2) | (6,6) | (6,6) | (6,6) |

In all experiments, we choose the number of moving and waiting vehicles to represent traffic characteristic $s_l^t$. To this order, first, for lane $l$ we consider a segment of 300 meters from the intersection and split it into three chunks of 100 meters. Then, $\alpha_{l,c}^t$ for $c = 1, 2, 3$ is the number of moving vehicles in chunk $c$ of the lane $l$ at time $t$. Also, we define $\beta_l^t$ as the number of waiting vehicles at lane $l$ at time $t$. Now, we represent the traffic characteristic of lane $l$ by $s_l^t := [\alpha_{l,1}^t, \alpha_{l,2}^t, \alpha_{l,3}^t, \beta_l^t]$.

It is assumed that the traffic always can turn to the right unless there is conflicting traffic or there is a "no turn on red" signal. To clear the intersection, the green light is followed by 5 seconds of yellow light. For each intersection, we run the planning for the next 600 seconds with a minimum active time of 10 seconds for each phase. To simulate the environment, we used CityFlow [36]. For more details on data construction, simulator, and intersections configurations, see Appendix.

### 5.1 Results of the Single-Environment Regime

In this experiment, we train the AttendLight for a particular intersection instance and then test it for the same intersection configuration. We compare AttendLight with SOTL [12], Max-pressure [28],

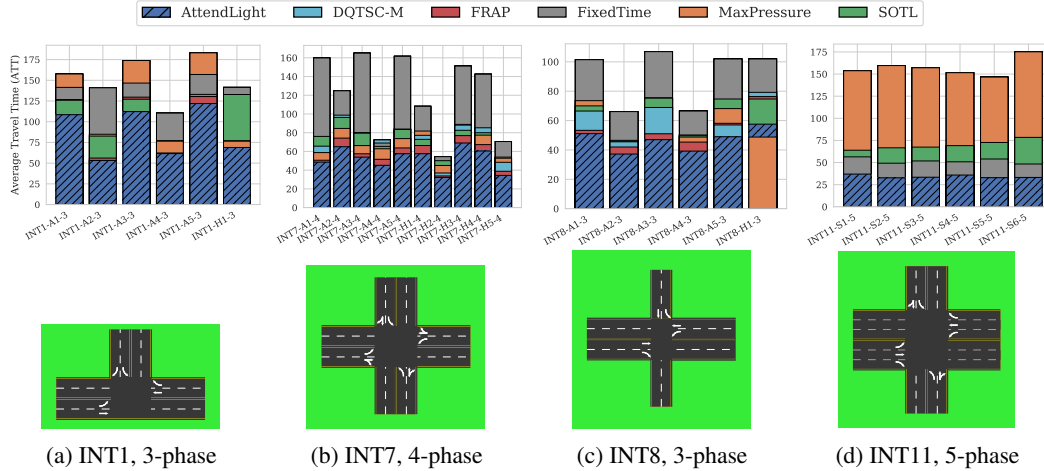

(a) INT1, 3-phase     (b) INT7, 4-phase     (c) INT8, 3-phase     (d) INT11, 5-phase

Figure 3: The comparison of AttendLight in single-env regime with baseline algorithms. The corresponding intersection configuration is depicted below the ATT results.

Fixed-time policies, DQTSC-M [24], and FRAP [37]. The FRAP by design does not handle all the intersections that we consider in this work, so we compare against it when possible. For the purpose of comparison, we measure the Average Travel Time (ATT) for all algorithms. We consider a variety of common real-world intersections configurations with {2,3,4,5,8} phases, {1,2,3} lanes, {3,4} ways, and one- and bi-directional intersections. For example, Figure 3 presents the results for four different cases. As it is shown, in most cases AttendLight outperforms benchmarks algorithms in terms of ATT, and FRAP is the second-best working algorithm. In Figure 3d, FRAP is not applicable. When considering all 112 cases, AttendLight yields 46%, 39%, 34%, 16%, and 9% improvement over FixedTime, MaxPressure, SOTL, DQTSC-M, and FRAP, respectively. Thus, when AttendLight is used solely to train a single environment, it works well for all the available number of roads, lanes, phases, and all traffic-data. Further results for other intersections are available in Appendix A.5.

## 5.2 Results of Multi-Environment Regime

We evaluate the key feature of AttendLight that enables it to be utilized for multiple TSCPs. We divide the set of intersection instances into two segments: training and testing sets, each with 42 and 70 instances, respectively. We train AttendLight by running all 42 training intersection instances in parallel to obtain data for the training and use the REINFORCE algorithm to optimize the trainable parameters. Once the trained model is available, we test the performance on training and testing instances. The testing set includes new intersection topologies as well as new traffic-data that has not been observed during training. Appendix C.2 summarizes the details of both sets.

To the best of our knowledge, there is no RL-based algorithm in the literature that works on more than one intersection instance without any transfer learning or retraining. Hence, to evaluate the performance of AttendLight in the multi-env regime, we have no choice other than comparing it against the single-env regime and the previously explained baselines, e.g., SOTL, MaxPressure, FixedTime, DQTSC-M, and FRAP. Recall that for the AttendLight in single-env regime, DQTSC-M, and FRAP, a separate model is trained for every instance. In contrast, the multi-env regime trains a single model to make the traffic decisions of multiple intersections. Clearly, having such a "general" model will lead to an inevitable performance loss, which makes the comparison unfair. Nevertheless, we embrace such unfairness in our experiments. Comparing AttendLight in multi-env versus single-env also allows measuring the performance degradation as a result of training a general model. Finally, to make sure that the results are robust against the randomness, we trained five models with different random seeds and always report the average statistics.

Now, we would like to evaluate the performance of the multi-env regime versus single-env. Our goal in this analysis is twofold: In case (I) we evaluate how introducing a universal model trained in multi-env regime exacerbates the ATT in the training set, and case (II) demonstrates how the

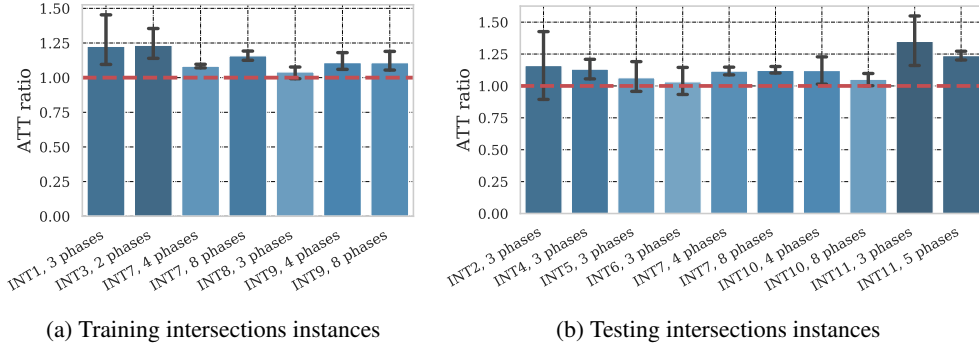

(a) Training intersections instances          (b) Testing intersections instances

Figure 4: The ATT of multi-env policy divided by ATT of single-env policy. The error bars represent 95% confidence interval for ATT ratio. The closer ATT ratio is to one (red dashed line), the less degradation is caused as a result of using a multi-env model.

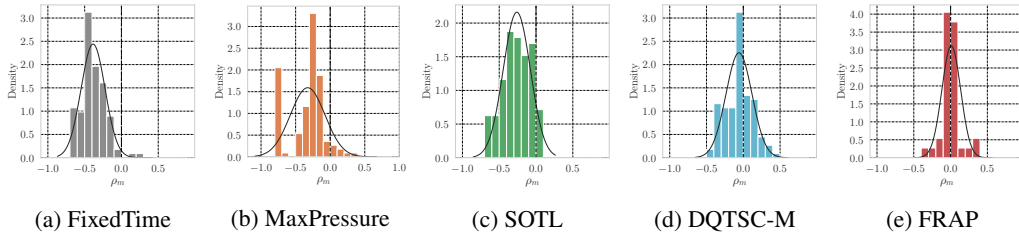

(a) FixedTime     (b) MaxPressure     (c) SOTL     (d) DQTSC-M     (e) FRAP

Figure 5: These plots illustrate the density of $\rho_m$ over all intersections. Here $\rho_m < 0$ means that the multi-env outperforms the baseline algorithm.

multi-env model generalizes to testing instances which are not visited during training. To measure this, we calculate *ATT ratio*, which is ATT in multi-env divided by ATT of the single-env regime. The closer ATT ratio is to one, the less degradation is caused as a result of using a multi-env model. Figure 4 summarizes the ATT ratio for all 112 intersection instances. Each bar in this plot illustrates the average ATT ratio for a number of traffic-data associated with an intersection configuration and the error bars represent the 95% confidence interval. To analyze case (I), we use Figure 4a. We observe that in most cases the results of the multi-env regime are close to that of the single-env regime. In particular, we have an average 15% ATT degradation with a standard deviation of 0.15. Case (II) analysis is demonstrated in Figure 4b. We observe that the trained policy works well in the majority of intersection/traffic-data such that on average it has 13% ATT gap with a standard deviation of 0.19. Furthermore, from this figure, one may notice that there are several intersection instances that the model-trained in the multi-env regime has lower ATT. We conjecture that such behavior is due to knowledge sharing between intersections, meaning that environments are sharing successful phase controls that are not necessarily explored in single-env training cases.

Next, we compare the multi-env regime with all the benchmark algorithms. To this end, we define $\rho_m = \frac{u_m - b_m}{\max(u_m, b_m)} \in [-1, 1], \forall m \in \mathcal{M}$, in which $u_m$ is the ATT of intersection $m$ when the trained model of multi-env is scored greedily (i.e., we let the phase with the highest probability to be the next active phase), and $b_m$ is the ATT of the corresponding intersection of a certain benchmark algorithm. Having $\rho_m < 0$ means that the multi-env model outperforms the baseline algorithm. The density plots of Figure 5 show the distribution of $\rho_m$ for all $m \in \mathcal{M}$ and for different baseline algorithms. Figures 5a-5c indicate that multi-env obtains smaller ATT when compared to MaxPressure, FixedTime, and SOTL. Noting that the fitted Normal distribution is centered at a negative value and only a small tail of the distribution lies on the positive side suggests that the multi-env model almost always has a smaller ATT. As Figure 5d suggests, there are few cases where DQTSC-M obtains smaller ATT; however, in average AttendLight multi-env regime achieves smaller ATT than DQTSC-M. Figure 5e shows that multi-env AttendLight provides competitive performance with respect to FRAP, which is trained on single-env setting.

In summary, we can conclude that the multi-env algorithm works well in the training set and also obtains quite a small gap in unseen environment instances. So, the model can be used without any retraining on new intersection instances similar to those observed during training.

### 5.3 Few-Shot Training for Calibration

Although the multi-env regime works well in practice, one might want to get a specialized policy for an important intersection. For this purpose, we start with the universal policy and after a few training steps, we obtain an improved policy for that intersection. Recall that AttendLight is designed to handle any intersection, so we do not need to modify the structure of the policy network (the number of inputs and outputs). Following this strategy, the maximum and average of the multi-env gap compared to single-env decreases significantly after 200 training episodes (instead of 100,000 episodes when trained from scratch) such that we got to 5% gap on average with respect to the single-env regime. After 1000 training steps this gap decreases to 3%. See Appendix A.7 for more details.

## 6   Conclusion and Discussion

In this paper, we consider the traffic signal control problem, and for the first time, we propose a universal RL model, called AttendLight, which is capable of providing efficient control for any type of intersections. To provide such capability to the model, we propose a framework including two attention mechanisms to make the input and output of the model, independent of the intersection structure. The experimental results on a variety of scenarios verify the effectiveness of the AttendLight. First, we consider the single-environment regime. In this case, AttendLight outperforms existing methods in the literature. Next, we consider AttendLight in the multi-environment regime in which we train it over a set of distinct intersections. The trained model is tested in new intersections verifying the generalizability of AttendLight.

A future line of research could be extending AttendLight to control multiple intersections in a connected network of intersections. Another research direction would be applying different RL algorithms such as Actor-Critic, A2C, and A3C to improve the numerical results. In a similar approach, considering other embedding functions and attention mechanisms would be of interest. In addition, the AttendLight framework is of independent interest and can be applied to a wide range of applications such as *Assemble-to-Order Systems, Dynamic Matching Problem,* and *Wireless Resource Allocation* problems. Similar to TSCP, each of these problems has to deal with the varying number of inputs and outputs, and AttendLight can be applied with slight modifications.

## Broader Impact

In this paper, the authors propose AttendLight, a Deep Reinforcement Learning algorithm to control the traffic signals efficiently and autonomously. Utilizing AttendLight for controlling traffic signals brings several benefits to society.

First, this algorithm is responsive to the dynamic behavior of traffic movement and provides a control policy to an intersection to minimize the travel time. This has several societal implications:

- *Less traffic*: according to information gathered in [9] in 2015, drivers in the United States wasted 6.9 billion hours annually in traffic. With AttendLight people will spend less time in traffic jams.
- *Lower fuel consumption*: it has been shown that about 3 billion gallons of gas wasted in 2014 due to traffic congestion [9]. AttendLight will help to reduce fuel consumption by easing traffic flows.
- *Cleaner environment*: it is predicted that air pollution will cause around 2.5 million cases of non-communicable disease by 2035, should the air quality stay the same as in 2018 [21]. People will breathe higher quality air if we have smarter traffic signal controllers.

Second, in contrast to previous RL models, AttendLight does not need to be trained for every new intersection. Thanks to the attention mechanism, indeed AttendLight is a universal model that can be simply deployed for any type of intersection after it is trained over a collection of distinct intersections. As long as the structure for the new intersection follows a similar distribution as the training set, AttendLight provides accurate results. This capability is the key advantage of this model because, designing a new model imposes several costs such as (i) human expertise, (ii) the required computational power, and (iii) data collection resources. Therefore, sparing experts from repetitive work as well as saving computational resources are other societal impacts of AttendLight.

One limitation of AttendLight is that it may fail to come up with an efficient policy whenever the intersection topology is very complex and unusual, such that the training set does not involve a similar structure. We hope that such a limitation can be addressed by increasing the training set diversity and incorporating more complex policy models.

Future researchers are encouraged to consider extending AttendLight to control multiple intersections in a coordinated manner. Given a network of intersections, traffic signals have significant impacts on each other. Thus, controlling every individual signal without considering this incorporation may exacerbate the whole traffic. Reaching the point that AttendLight could control a network of intersections while it respects the association effects, we expect to achieve too many other valuable societal impacts. However, the extension of AttendLight to the multi-intersection scenario is not quite straightforward. Specifically, one needs to consider several challenges such as scalability of the proposed model, and how to involve the coordination in decision-making procedure. See more details in [20].

Further motivation to pursue this research would be improving the RL algorithm used to train the AttendLight model. In the current research, we utilized a policy-gradient RL algorithm called REINFORCE. Despite the superior numerical results of AttendLight, there is definitely room for improvement. As a low-hanging fruit, other state-of-the-art policy-based RL algorithms such as Actor-Critic, A2C, and A3C can substitute REINFORCE in AttendLight.

## Acknowledgment

This work was fully supported by SAS Institute, Inc. During the time to conduct the research all authors were full time employees of SAS Institute, Inc, and used the resources and computing power of SAS Institute, Inc.

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
