[Supplementary Material]

# Supplementary Material for "AttendLight: Universal Attention-Based Reinforcement Learning Model for Traffic Signal Control"

## A    Numerical Experiments Details

Appendix A includes the details of all numerical experiments. In Appendix A.1, we describe our traffic data and how we generate the synthetic data. Appendices A.2 and A.3 explains the AttendLight model as well as the RL training that we consider throughout the paper. Appendix A.4 gives a brief explanation on our baseline algorithms. Finally, in Appendix A.5, we provide extensive results of AttendLight for single-env and multi-env regimes.

### A.1    Details of Real-World Data and the Synthetic Data Generation

In each intersection, we consider three traffic movement sets, namely straight, turn-left, and turn-right.These sets are used in defining the traffic data of an intersection in both real-world and synthetic cases. Let us reconsider intersection of Figure 1 and the corresponding traffic movements in Figure 1a. For this intersection, the straight, turn-right, and turn-left sets are $\{\nu_2, \nu_3\}$, $\{\nu_4, \nu_6\}$, and $\{\nu_1, \nu_5\}$, respectively. Next, we elaborate two sources of traffic that we use in our experiments.

**Real-world traffic data**
For the real-world data, we use the public data of intersections in Hangzhou and Atlanta [33, 37], which are available to download in [2] and [1], respectively. These data-sets include the traffic on 4-way intersections with straight and turn-left traffic movements. For 3-way intersections, we slightly modify the data of 4-way intersections from Hangzhou and Atlanta to adapt them for 3-way structures. In particular, for each vehicle arrival, we keep the vehicle properties and the arrival time, but we change the entering and leaving lanes if they are not available in the new intersection. To do this, we allocate the traffic of unavailable movement to an available one from the same set (e.g., straight or turn-left) uniformly at random. For example, consider intersection of Figure 1. Compared to a 4-way intersection, this intersection does not have north-to-south and south-to-north traffic movements. So, to use traffic data A1-A5 and H1-H5, we substitute the traffic movements north-to-south and south-to-north with the traffic movements east-to-west and west-to-east. We follow the same process for the left-turn traffic movements.

**Synthetic traffic data**
For the intersections where there exist an entering/leaving roads with three-lanes, we could not find any real-world data of a single intersection with the right-turn data. So, we generate traffic-data with a different rate of incoming vehicles. For this purpose, we consider two Poisson arrival processes with a rate $\lambda$ from $\{3, 4\}$ (seconds). Incoming traffic with the probability of 70%, 20%, and 10% is selected to go straight, turn-left, or turn-right, respectively. Inside each of three sets (i.e., straight, turn-left, and turn-right sets), the traffic movements are selected uniformly. For example, consider the intersection of Figure 1, and assume a vehicle is chosen to move straight. Then, with the probability of 50% either $\nu_2$ or $\nu_3$ will be selected. In addition, to analyze the performance of each algorithm under the different volumes of traffic, we increase the number of arriving vehicles in three settings: at each arrival time, an additional vehicle is added with a probability $p$ from $\{0.05, 0.1, 0.3\}$. Overall, there are six cases of synthetic traffic data, which are summarized in Table 2.

Table 2: Parameters of the synthetic data. Each parenthesis shows $\lambda$ of the Poisson distribution and the probability of having two vehicle arrival at each time.

| traffic-data | S1 | S2 | S3 | S4 | S5 | S6 |
|---|---|---|---|---|---|---|
| $(\lambda, p)$ | (4, 0.3) | (4, 0.1) | (3, 0.1) | (3, 0.05) | (3, 0.3) | (4, 0.05) |

## A.2 Details of AttendLight Policy Model

The AttendLight policy model—visualized in Figure 2—contains five trainable components: embedding layer, `state-attention`, Recurrent Neural Network (RNN), fully-connected layer, and `action-attention`. In this work, we use a 1-dimensional convolution layer with the input-channel of size four, kernel-size and stride of size one, and the out-channel of size $d$. This embedding is an affine transformation that maps all lane characteristics to a $d$-dimensional space for allowing more effective feature representations. The `state-attention` uses linear transformations of query and references with output size of $d$. For the RNN, we use a single layer of LSTM cell with a hidden dimension $d$. The output of the LSTM is passed into a linear layer with output size $d$ followed by a ReLU activation function. The `action-attention` model also uses the same linear transformations for the query and references with $d$ output dimension and returns the probability of selecting each action. To train the single-env models, we set $d = 128$ and the multi-env model use $d = 256$. We would like to emphasize that we have not done any structured hyper-parameter tuning of AttendLight. For training the AttendLight policy model, we use a policy-gradient based method as explained next.

## A.3 Details of the RL Training

We use a variance-reduced variant of the REINFORCE algorithm [27] to train the AttendLight. The details of this algorithm are presented in Algorithm 1. In REINFORCE, there are two neural networks, called the actor (with parameters $\theta$) and critic (with parameter $\phi$). The actor network is responsible for learning the optimal action, while the critic is used for variance reduction. The critic uses the output of the `state-attention` which is followed by two fully-connected layers with ReLU activation function and returns the value of being in a state.

We train the single-env models for 100000 episodes and the multi-env models for 30000 episodes. To train these models, we employ the Monte-Carlo simulation using the current policy $\pi_\theta$ to produce a valid sequence of state, action, and rewards. Given the sequence of observations, we run a single train-step of the REINFORCE algorithm and update the weights of the networks. We use Adam optimizer with the learning rate of 0.005 for the single-env regime and 0.0005 for the multi-env regime. To stabilize the training of the single-env regime, we consider $n = 3$ instances of the intersection in parallel. For this problem, every 1100 episodes take approximately one hour on a single V100 GPU. On the multi-env regime, we choose $n = 42$ environment instances (without replacement) and run them simultaneously. Not surprisingly, the multi-env regime takes a longer time such that it accomplishes 85 episodes in one hour on the same GPU.

---

**Algorithm 1:** REINFORCE Algorithm

1:   initialize the actor and critic networks with random weights $\theta$ and $\phi$, respectively
2:   **for** $iteration = 1, 2, \cdots$ **do**
3:      reset gradients: $d\theta \leftarrow 0, d\phi \leftarrow 0$
4:      if single-env regime, get $n$ copy of the same environment instances; in multi-env regime choose randomly $n$ environment instances
5:      **for** $i = 1, \cdots, n$ **do**
6:         initialize step counter $t \leftarrow 0$
7:         **repeat**
8:            choose *next-action* $a_i^t$ according to the distribution $\pi(\cdot|s_i^t; \theta)$
9:            observe new state $s_i^{t+1}$ and reward $r_i^t$ which is the negative of intersection pressure
10:           $t \leftarrow t + 1$
11:         **until** termination condition $t \leq T$ is not satisfied
12:      **end for**
13:      compute the cumulative reward $R_i^t = \sum_{t'=t}^{T} r_i^{t'}$ for all $i = 1, \cdots, n$ and $t = 1, \cdots, T$
14:      $d\theta \leftarrow \frac{1}{nT} \sum_{i=1}^{n} \sum_{t=1}^{T} \left(R_i^t - V(s_i^t; \phi)\right) \nabla_\theta \log \pi(a_i^t|s_i^t; \theta)$
15:      $d\phi \leftarrow \frac{1}{nT} \sum_{i=1}^{n} \sum_{t=1}^{T} \nabla_\phi \left(R_i^t - V(s_i^t; \phi)\right)^2$
16:      update $\theta$ using $d\theta$ and $\phi$ using $d\phi$.
17: **end for**

---

### A.4 Details of Baseline Algorithms

#### A.4.1 SOTL

Let's define counters $\alpha$ and $\beta$ to count the number of cars behind the phases with the red signal and the number of cars behind the phase with the green signal, respectively. In SOTL algorithm [12], if the active time of the current phase is more than a threshold $\delta$, and if $\alpha >$ `max-red-count` the $\beta <$ `min-green-count`, then the active phase switches in a cyclic manner. To obtain the best values for the parameters of the SOTL, we try a grid-search over set $\{2, 7, \ldots, 62\}$ for `max-red-count` and `min-green-count`, $\{2, 7, \ldots, 33\}$ for $\delta$. This results in $13 \times 13 \times 7 = 1183$ cases, in which we report the minimum ATT obtained among these cases.

#### A.4.2 Fixed-Time

In the Fixed-Time algorithm, the phases change in a cyclic order. For each phase, an active time is determined in advance, and once the active phase meets that active time, the phase switches to the next one. This is the most common approach in real-world intersections. We set the active time of each phase to 15 seconds.

#### A.4.3 Max-Pressure

The Max-Pressure algorithm [28] considers the pressure of participating lanes for each phase, and it activates the phase with the highest pressure for the next time-step.

#### A.4.4 FRAP

The get the result of the FRAP algorithm [37], we run the publicly available code of the algorithm for our intersection instances. FRAP has two main limitations which make it inapplicable to some of the intersections that we consider: (i) it assumes that each phase involves exactly two traffic movements; and (ii) it is designed based on the assumption that the number of traffic movements in all phases are the same. The first limitation is due to the implementation of algorithm [3]. We adjusted the implementation by generating dummy traffic movements to handle intersections such as `INT1` and `INT2`. However, the second limitation is due to the design of FRAP. This is because, FRAP obtains an embedding of the traffic characteristic ($f_p^\nu$) and the phase-id ($f_p^s$) for phase $p$ by:

$$
\begin{aligned}
h_p^\nu &= \text{RELU}(W^\nu f_p^\nu + b^\nu), \\
h_p^s &= \text{RELU}(W^s f_p^s + b^s),
\end{aligned}
\tag{4}
$$

in which $W^\nu$, $W^s$, $b^\nu$, and $b^s$ and trainable variables, $h_p^\nu$ and $h_p^s$ are the embedded traffic characteristic and the phase. Then, these are combined with adding another layer of affine transformation to obtain the state definition for each phase:

$$
h_p^\nu = \text{RELU}(W^h[h_p^\nu, h_p^s] + b^h).
\tag{5}
$$

Since $W^\nu$ is shared among all phases, it is not applicable when phases have a different number of lanes. So, it is not possible to adjust the algorithm to work for some of the intersections such as `INT3` and `INT11`.

#### A.4.5 DQTSC-M

Since there was not any publicly available code of DQTSC-M [24], we implemented the algorithm with some slight modifications. We used CityFlow [36] to get the state and reward of each time-step. We implemented the DQN [18] algorithm as is described in the paper. Three convolutional layers with filter sizes $2 \times 4$, $2 \times 4$, and $2 \times 2$, with stride of $1 \times 2$, $1 \times 2$, and $1 \times 3$ are used in the layer one, two, and three, respectively. The output of the convolution layers is passed into a fully connected layer with 128 nodes, followed by another fully connected layer of 64 nodes and the last layer provides the Q-value of each possible action.

### A.5 Extended Comparison Results

Tables 3, 4, and 5 show the average travel time (ATT) for all environment instances for FixedTime, MaxPressure, SOTL, FRAP, and AttendLight in single-env and multi-env regimes. When FRAP is

not applicable, we report "NA" in these tables. As it is shown, AttendLight in single-env regime outperforms other algorithms in 107 cases out of 112 cases and it obtains 46%, 39%, 34%, 16%, and 9% average ATT improvement over FixedTime, MaxPressure, SOTL, DQTSC-M, and FRAP, respectively. In addition, Figure 6 illustrates the ATT comparisons results for all intersections. Furthermore, the result of the multi-env regime is also reported under which AttendLight outperforms other benchmarks in 57 cases and it achieves 39%, 32%, 26%, and 5% improvement over FixedTime, MaxPressure, SOTL, and DQTSC-M, respectively. In comparison to FRAP, AttendLight in multi-env obtains 3% larger ATT in average.

Table 3: Results of all algorithms for `INT1`–`INT5`

| case | FixedTime | MaxPressure | SOTL | FRAP | DQTSC-M | AttendLight single-env | AttendLight multi-env |
|---|---|---|---|---|---|---|---|
| `INT1, 3 phases` | | | | | | | |
| `INT1-A1-3` | 141.44 | 157.59 | 125.93 | 126.73 | 124.34 | **108.47** | 122.61 |
| `INT1-A2-3` | 140.96 | 84.86 | 82.65 | 56.12 | 76.74 | **53.27** | 62.25 |
| `INT1-A3-3` | 146.65 | 173.89 | 127.2 | 129.66 | 129.00 | **112.23** | 124.87 |
| `INT1-A4-3` | 110.4 | 76.51 | 77.18 | 62.28 | 75.52 | **61.53** | 67.69 |
| `INT1-A5-3` | 157.02 | 183.15 | 132.98 | 130.41 | 139.23 | **121.94** | 129.79 |
| `INT1-H1-3` | 141.44 | 76.51 | 132.88 | 77.04 | 77.04 | **68.92** | 122.61 |
| `INT2, 3 phases` | | | | | | | |
| `INT2-A1-3` | 99.64 | 68.3 | 76.66 | **42.48** | 54.40 | 43.35 | 68.32 |
| `INT2-A3-3` | 100.44 | 71.16 | 70.27 | 50.27 | 57.78 | **46.67** | 66.46 |
| `INT2-A4-3` | 76.86 | 47.36 | **31.44** | 39.49 | 45.08 | 35.87 | 30.32 |
| `INT2-A5-3` | 95.14 | 86.26 | **42.29** | 48.32 | 60.58 | 52.29 | 43.19 |
| `INT2-H1-3` | 76.86 | 63.67 | 46.63 | 42.24 | 45.23 | **39.04** | 44.07 |
| `INT3, 3 phases` | | | | | | | |
| `INT3-A1-2` | 102.39 | 99.34 | 80.49 | NA | 89.49 | **63.75** | 81.39 |
| `INT3-A2-2` | 71.76 | 65.47 | 51.46 | NA | 60.20 | **45.37** | 51.21 |
| `INT3-A3-2` | 109.97 | 115.05 | 82.97 | NA | 86.78 | **71.99** | 81.81 |
| `INT3-A4-2` | 68.78 | 58.44 | 43.09 | NA | 48.06 | **39.45** | 43.42 |
| `INT3-A5-2` | 109.14 | 107.65 | 80.06 | NA | 85.10 | **66.83** | 83.42 |
| `INT3-H1-2` | 102.41 | 58.44 | 80.41 | NA | 62.35 | **53.81** | 81.24 |
| `INT4, 3 phases` | | | | | | | |
| `INT4-A1-3` | 111.27 | 72.33 | 73.04 | NA | 49.75 | **41.81** | 59.36 |
| `INT4-A3-3` | 99.64 | 77.47 | 76.35 | NA | 57.56 | **46.46** | 53.64 |
| `INT4-A4-3` | 95.14 | 54.76 | **30.19** | NA | 45.14 | 33.47 | 30.58 |
| `INT4-A5-3` | 70.38 | 73.99 | 46.47 | NA | 59.17 | 43.75 | **41.34** |
| `INT4-H1-3` | 72.24 | 56.4 | 47.23 | NA | 46.12 | **36.06** | 47.17 |
| `INT4-S1-3` | 56.96 | 44.88 | 61.32 | NA | 35.67 | **32.42** | 36.23 |
| `INT4-S2-3` | 82.61 | 45.69 | 59.91 | NA | 32.74 | **30.9** | 33.93 |
| `INT4-S3-3` | 52.13 | 52.06 | 46.37 | NA | 34.53 | **32.68** | 35.78 |
| `INT4-S4-3` | 85.47 | 55.4 | 49.2 | NA | 34.48 | **31.8** | 37.04 |
| `INT4-S5-3` | 70.09 | 64.61 | 52.64 | NA | 38.18 | **34.96** | 40.1 |
| `INT4-S6-3` | 87.47 | 47.19 | 72.3 | NA | 33.06 | **31.07** | 33.68 |
| `INT5, 3 phases` | | | | | | | |
| `INT5-A2-3` | 70.38 | 46.88 | 45.88 | NA | 46.25 | 35.76 | **29.35** |
| `INT5-A3-3` | 99.64 | 64.57 | 68.75 | NA | 56.21 | **48.11** | 63.64 |
| `INT5-A4-3` | 111.27 | 71.13 | 71.87 | NA | 62.26 | 51.46 | **39.67** |
| `INT5-A5-3` | 100.44 | 63.28 | 67.94 | NA | 49.26 | **47.38** | 70.52 |
| `INT5-H1-3` | 72.24 | 48.04 | 43.63 | NA | 46.61 | **38.2** | 40.75 |
| `INT5-S1-3` | 40.42 | 37.84 | 32.31 | NA | 29.11 | **28.74** | 29.62 |
| `INT5-S2-3` | 43.83 | 37.23 | 33.07 | NA | 29.46 | **28.41** | 29.96 |
| `INT5-S3-3` | 50.79 | 37.57 | 33.68 | NA | 30.60 | **28.82** | 29.75 |
| `INT5-S4-3` | 53.71 | 39.12 | 33.34 | NA | 30.52 | **28.21** | 29.46 |
| `INT5-S5-3` | 62.29 | 43.00 | 34.43 | NA | 31.43 | **30.27** | 31.52 |
| `INT5-S6-3` | 45.18 | 37.69 | 33.75 | NA | 29.14 | **28.37** | 29.3 |

## A.6 Mean State Results

The proposed AttendLight framework uses the state-attention to produce a unified state representation from a varying number of lane-traffic information. The state attention learns the importance of each lane-traffic $g(s_l^t)$ and the weighted sum of them creates the phase-state $z_l^t$. Instead, one could simply use the average operator instead of the state-attention. In other words, just use equal weight $\frac{1}{|\mathcal{L}_p|}$ for all participating lane-traffic $g(s_l^t)$ in phase $p$.

Figure 6: The ATT comparison result of all baseline algorithms with AttendLight in single-env regime. Note that FRAP is not applicable to some of the intersections.

Table 4: Results of all algorithms for `INT6-INT9`

| case | FixedTime | MaxPressure | SOTL | FRAP | DQTSC-M | AttendLight single-env | AttendLight multi-env |
|---|---|---|---|---|---|---|---|
| `INT6, 3 phases` | | | | | | | |
| `INT6-A2-3` | 70.38 | 49.33 | 46.2 | NA | 42.97 | 35.29 | **29.01** |
| `INT6-A3-3` | 105.64 | 67.97 | 81.53 | NA | 60.91 | **58.94** | 70.13 |
| `INT6-A4-3` | 117.38 | 76.04 | 83.04 | NA | 68.77 | 55.07 | **38.69** |
| `INT6-A5-3` | 105.27 | 65.66 | 72.73 | NA | 59.08 | **51.92** | 76.29 |
| `INT6-H1-3` | 72.18 | 54.11 | 43.43 | NA | 41.70 | 39.77 | **39.57** |
| `INT6-S1-3` | 40.36 | 37.06 | 32.27 | NA | 29.21 | **28.85** | 29.66 |
| `INT6-S2-3` | 43.82 | 37.22 | 33.02 | NA | 29.60 | **28.51** | 29.58 |
| `INT6-S3-3` | 50.78 | 37.54 | 33.64 | NA | 31.25 | **28.88** | 29.44 |
| `INT6-S4-3` | 53.71 | 39.11 | 33.32 | NA | 30.42 | **28.39** | 29.27 |
| `INT6-S5-3` | 62.27 | 42.95 | 33.46 | NA | 32.78 | **30.26** | 31.5 |
| `INT6-S6-3` | 45.14 | 37.64 | 33.71 | NA | 29.18 | **28.47** | 29.07 |
| `INT7, 4 phases` | | | | | | | |
| `INT7-A1-4` | 159.82 | 58.76 | 76.05 | 50.6 | 65.62 | **48.56** | 51.98 |
| `INT7-A2-4` | 124.78 | 84.57 | 96.34 | 74.24 | 98.75 | **65.01** | 69.12 |
| `INT7-A3-4` | 164.97 | 66.26 | 79.49 | 57.49 | 80.13 | **53.91** | 59.11 |
| `INT7-H4-4` | 142.46 | 77.39 | 80.00 | 67.28 | 85.26 | **60.47** | 70.55 |
| `INT7-A4-4` | 68.97 | 62.86 | 65.41 | 51.61 | 72.32 | **45.18** | 50.07 |
| `INT7-A5-4` | 161.59 | 73.72 | 83.49 | 63.72 | 83.87 | **57.53** | 62.32 |
| `INT7-H5-4` | 70.59 | 52.78 | 53.95 | 38.81 | 48.40 | **34.38** | 37.43 |
| `INT7-H1-4` | 108.17 | 81.77 | 72.9 | 66.32 | 77.21 | **57.57** | 61.82 |
| `INT7-H3-4` | 151.16 | 88.88 | 82.58 | 76.9 | 88.10 | **69.01** | 77.5 |
| `INT7-H2-4` | 54.43 | 45.01 | 50.13 | 33.81 | 36.99 | **32.26** | 35.04 |
| `INT7, 8 phases` | | | | | | | |
| `INT7-A1-8` | 159.82 | 68.47 | 84.35 | 49.82 | 94.24 | **49.11** | 59.55 |
| `INT7-A2-8` | 124.78 | 50.72 | 61.74 | 40.14 | 73.17 | **35.81** | 40.57 |
| `INT7-A3-8` | 164.97 | 81.06 | 86.75 | 57.11 | 78.46 | **53.03** | 58.12 |
| `INT7-A4-8` | 68.97 | 54.46 | 66.25 | 50.94 | 76.97 | **42.4** | 50.22 |
| `INT7-A5-8` | 161.59 | 82.23 | 88.12 | 62.96 | 65.63 | **55.31** | 62.23 |
| `INT7-H1-8` | 108.17 | 93.37 | 80.83 | 73.14 | 80.04 | **52.25** | 62.41 |
| `INT7-H2-8` | 54.43 | 42.37 | 53.09 | 32.56 | 35.42 | **31.03** | 34.03 |
| `INT7-H3-8` | 151.16 | 90.83 | 98.84 | 72.18 | 103.61 | **66.59** | 77.7 |
| `INT7-H4-8` | 142.46 | 75.08 | 84.15 | 61.59 | 91.98 | **55.8** | 62.2 |
| `INT7-H5-8` | 70.59 | 42.66 | 56.4 | 34.77 | 45.43 | **32.19** | 35.7 |
| `INT8, 3 phases` | | | | | | | |
| `INT8-A1-3` | 101.44 | 73.55 | 70.00 | 53.34 | 66.48 | **51.09** | 53.09 |
| `INT8-A2-3` | 66.1 | 45.87 | 46.53 | 42.00 | 45.53 | **37.19** | 40.84 |
| `INT8-A3-3` | 106.93 | 75.53 | 75.26 | 51.00 | 68.91 | **46.99** | 49.37 |
| `INT8-A4-3` | 66.48 | 48.77 | 49.7 | 45.34 | 50.20 | **39.2** | 41.57 |
| `INT8-A5-3` | 101.92 | 68.15 | 74.68 | 58.13 | 57.01 | **48.97** | 52.7 |
| `INT8-H1-3` | 101.92 | **48.77** | 74.68 | 76.33 | 79.21 | 57.54 | 53.09 |
| `INT9, 4 phases` | | | | | | | |
| `INT9-S1-4` | 66.56 | 55.42 | 52.69 | NA | 41.69 | **36.4** | 40.72 |
| `INT9-S2-4` | 57.51 | 51.45 | 52.94 | NA | 36.73 | **33.49** | 35.34 |
| `INT9-S3-4` | 61.41 | 52.08 | 51.28 | NA | 39.84 | **32.72** | 35.38 |
| `INT9-S4-4` | 56.24 | 53.4 | 55.16 | NA | 39.26 | **34.56** | 36.42 |
| `INT9-S5-4` | 56.09 | 48.21 | 55.37 | NA | 35.66 | **32.24** | 34.16 |
| `INT9-S6-4` | 56.09 | 53.4 | 55.37 | NA | 35.17 | **31.77** | 40.72 |
| `INT9, 8 phases` | | | | | | | |
| `INT9-S1-8` | 66.56 | 54.54 | 59.87 | NA | 52.70 | **35.95** | 39.87 |
| `INT9-S2-8` | 57.51 | 47.71 | 59.15 | NA | 34.26 | **32.08** | 33.55 |
| `INT9-S3-8` | 61.41 | 50.32 | 57.63 | NA | 37.42 | **32.07** | 34.43 |
| `INT9-S4-8` | 56.24 | 46.21 | 61.97 | NA | 38.44 | **33.05** | 35.3 |
| `INT9-S5-8` | 56.09 | 44.01 | 59.55 | NA | 34.60 | **31.21** | 32.75 |
| `INT9-S6-8` | 56.09 | 46.21 | 59.55 | NA | 33.69 | **30.63** | 39.87 |

Table 5: Results of all algorithms for `INT10-INT11`

| case | FixedTime | MaxPressure | SOTL | FRAP | DQTSC-M | AttendLight single-env | AttendLight multi-env |
|---|---|---|---|---|---|---|---|
| `INT10, 4 phases` | | | | | | | |
| `INT10-S1-4` | 79.09 | 160.14 | 80.04 | NA | 44.30 | 41.21 | **37.46** |
| `INT10-S2-4` | 61.47 | 162.98 | 89.5 | NA | 38.49 | **34.91** | 44.76 |
| `INT10-S3-4` | 74.28 | 162.46 | 97.49 | NA | 40.64 | **35.37** | 38.04 |
| `INT10-S4-4` | 65.56 | 156.16 | 97.07 | NA | 42.81 | **37.46** | 38.22 |
| `INT10-S5-4` | 64.5 | 150.15 | 102 | NA | 36.99 | **33.44** | 38.48 |
| `INT10-S6-4` | 60.75 | 176.95 | 90.06 | NA | 37.37 | **34.03** | 43.85 |
| `INT10, 8 phases` | | | | | | | |
| `INT10-S1-8` | 47.54 | 158.52 | 66.52 | NA | 53.07 | 39.59 | **36.74** |
| `INT10-S2-8` | 61.47 | 161.62 | 89.5 | NA | 36.78 | **33.79** | 35.89 |
| `INT10-S3-8` | 65.56 | 162 | 97.07 | NA | 38.91 | **35.78** | 38.49 |
| `INT10-S4-8` | 74.28 | 154.2 | 97.49 | NA | 41.14 | **36.14** | 37.1 |
| `INT10-S5-8` | 64.5 | 149.01 | 102 | NA | 38.63 | **32.73** | 37.03 |
| `INT10-S6-8` | 67.43 | 176.89 | 95.14 | NA | 36.56 | **32.52** | 35.41 |
| `INT11, 3 phases` | | | | | | | |
| `INT11-S1-3` | 47.54 | 153.77 | 66.52 | NA | 40.18 | **37.86** | 41.82 |
| `INT11-S2-3` | 45.04 | 159.59 | 70.25 | NA | 34.98 | **32.97** | 58.37 |
| `INT11-S3-3` | 49.24 | 157.1 | 87.33 | NA | 35.51 | **33.69** | 46.7 |
| `INT11-S4-3` | 47.2 | 151.54 | 81.91 | NA | 37.67 | **36.31** | 39.18 |
| `INT11-S5-3` | 47.76 | 146.58 | 88.9 | NA | 36.01 | **34.04** | 43.07 |
| `INT11-S6-3` | 45.03 | 175.13 | 85.71 | NA | 35.48 | **34.3** | 51.09 |
| `INT11, 5 phases` | | | | | | | |
| `INT11-S1-5` | 56.44 | 153.77 | 63.93 | NA | 39.47 | **36.98** | 44.02 |
| `INT11-S2-5` | 49.27 | 159.59 | 66.66 | NA | 32.79 | **32.72** | 42.14 |
| `INT11-S3-5` | 51.81 | 157.1 | 67.46 | NA | 35.51 | **33.43** | 43.7 |
| `INT11-S4-5` | 50.84 | 151.54 | 69.12 | NA | 38.51 | **35.88** | 44.25 |
| `INT11-S5-5` | 54.01 | 146.58 | 72.69 | NA | 35.60 | **32.89** | 40.29 |
| `INT11-S6-5` | 48.35 | 175.13 | 78.44 | NA | 35.02 | **33.18** | 39.24 |

Figure 7: The comparison of state-attention with mean-state. The $\rho$ distribution shows the ATT of mean-state over state-attention.

To analyze the performance of this approach, we modified the AttendLight framework accordingly, i.e., used the average operator instead of the state-attention and re-trained the single-env model for all 112 environment instances. Figure 7 shows the distribution of $\rho$ for the ATT ratio of mean-state over the attention-state. We can observe that the average is on the positive side, meaning that the ATT of mean-state are bigger than those in with the state-attention. Therefore, utilizing the state-attention model helps to achieve smaller ATT.

(a) Before fine-tuning     (b) 200-episode fine-tuning     (c) 1000-episode fine-tuning

Figure 8: These plots show the effect of fine-tuning for 200 and 1000 episodes, vs the multi-env without any fine-tuning.

### A.7 Few-Shot Training for Multi-Env Regime

In the multi-env regime, we can run a few-shot training to obtain a specialized policy for a certain intersection. To this end, we start with the trained multi-env policy and calibrate the policy for a specific intersection through a few training steps.

We implemented this approach and fine-tuned a policy for each of the 112 environment instances. Compared to the result of the single-env regime, the maximum and the average of multi-env gap decrease significantly after 200 training episodes (instead of 100,000 episodes when trained from scratch in single-env regime) such that the ATT-gap decreased to 5% gap in average (without fine-tuning it was 13%). After 1000 training steps, this gap decreased to 3%. Figure 8 shows the distribution of $\rho_m$ which defines $\rho_m$ by the single-env after fine-tuned and the single-env before the find tuning. As it is shown, after 200-episode of fine-tuning the distribution of $\rho$ is concentrated close to zero with a small standard deviation, leaning a bit toward the positive values. With 1000-episode, the distribution is leaned more toward zero with a smaller standard deviation. Need to mention that the fine-tuning is quite fast such that it takes 10 and 43 minutes to fine-tune the policy with 200 and 1000 episodes, respectively.

### A.8 Separated Train/Test ATT ratio distribution

In page 8, Figure 5 shows the distribution of $\rho_m$ for multi-env regime and all the baselines, for all 112 cases. To analyze the performance on test instances, we also depicted the plots for separated test and train environment instances. As it is shown in Figure 9, compared to the distribution of the train-instances, the distribution of the test-instances is leaned a bit toward the right side, which complies the same observation in Figure 4. Also, Figure 10 shows the distribution of $\rho_m$ when all baselines are compared with the multi-env policy which is improved by the few-shot training. As it is shown, in all the baselines the results improved such that the distribution is moved to the left and the variance of the distributions are smaller that those in Figure 9.

## B    Stochastic Training Regime

In this section, we propose an alternative multi-env regime, which we refer to it as *stochastic multi-env* regime, that accelerates the training when we have a large number of intersections. Since running Monte-Carlo simulation for the TSCP is quite costly, it might not feasible to incorporate all training intersection instances at each training step. Unlike the multi-env regime that we considered in this paper, the stochastic multi-env regime chooses a small mini-batch of the environment instances $n \ll |\mathcal{M}|$ to reduce the per iteration cost. The rest of the AttendLight algorithm is the same as the multi-env regime.

To test this approach, we randomly choose $n = 5$ environment instances from Table 6, and trained the AttendLight with a single layer LSTM of hidden dimension 256, using the learning rate 0.005 in

(a) Fixed-time    (b) MaxPressure    (c) SOTL    (d) DQTSC-M    (e) FRAP

(f) Fixed-time    (g) MaxPressure    (h) SOTL    (i) DQTSC-M    (j) FRAP

Figure 9: These plots illustrate the density of $\rho_m$ over train and test instances. The first row shows the distribution of the training intersection instances and the second row shows the testing intersection instances.

(a) Fixed-time    (b) MaxPressure    (c) SOTL    (d) DQTSC-M    (e) FRAP

(f) Fixed-time    (g) MaxPressure    (h) SOTL    (i) DQTSC-M    (j) FRAP

Figure 10: These plots illustrate the density of $\rho_m$ over train and test instances. The first row shows the distribution of the train-env instances and the second row shows the test-env instances.

Adam optimizer. Not surprisingly, this approach runs faster than multi-env regime such that it runs 385 episodes at each hour, 4.5 times faster than multi-env regime.

Figures 11a and 11b show the ATT on stochastic multi-env divided by the ATT on multi-env regime, and the 95% confidence interval among all traffic data of an intersection. We observe that most of the results (except two cases) are statistically equal. To do statistical analysis, the paired T-test on the hypothesis of equality of two regimes indicates that the average of means are statistically different at confidence level of 95% for INT3 and INT8. In addition, at 90% confidence level, the mean of two algorithms are statistically different for INT4 and INT11-3-phase.

In addition, Figures 11c and 11d show the ATT ratio for stochastic multi-env regime over the ATT of single-env regime. Considering the environment instances in the training set (Figure 11c), there is an average of 16% ATT degradation with the standard deviation of 0.15 (it was average of 15% and standard deviation of 15% in the multi-env regime). Similarly, on the test set (Figure 11d) stochastic multi-env regime has 15% ATT gap with the standard deviation of 0.22 (on multi-env regime it was 13% and standard deviation of 19%).

(a) Stochastic multi-env vs multi-env regime on training intersections instances

(b) Stochastic multi-env vs multi-env regime on testing intersections instances

(c) Stochastic multi-env vs single-env regime on training intersections instances

(d) Stochastic multi-env vs single-env regime on testing intersections instances

Figure 11: The ATT of stochastic multi-env policy divided by ATT of multi-env in sub-Figures a and b, and divided by ATT of single-env policy in sub-Figures c and d. The error bars represent 95% confidence interval for ATT ratio. The closer ATT ratio is to one (red dashed line), the less degradation is caused as a result of using a stochastic multi-env model.

(a) FixedTime    (b) MaxPressure    (c) SOTL    (d) DQTSC-M    (e) FRAP

Figure 12: These plots illustrate the density of $\rho_m$ over all intersections. Here $\rho_m < 0$ means that the stochastic multi-env outperforms the baseline algorithm.

Compared to the benchmark algorithms, in average it achieves 36%, 32%, 27%, and 2% improvement over FixedTime, MaxPressure, SOTL, and DQTSC-M algorithms, respectively. Similar to the multi-env regime, FRAP gets smaller ATT, which here is 7%. In addition, Figure 12 shows the distribution of $\rho_m$ for all environment instances, suggesting that in most cases stochastic multi-env regime obtains smaller ATT compared to FixedTime, MaxPressure, and SOTL algorithms.

In summary, stochastic multi-env works well and in most cases obtains statistically equal results with multi-env regime. So, if the number of cases in the training set goes beyond the power available computation resources, the stochastic multi-env regime is a reliable substitute.

## C    Environment Details

In these section, we provide the visualization of 11 intersections and details of the 112 environment instances.

**C.1 Intersection and Phase Visualizations**

Figure 13 and 14 show the visualization of all 11 intersections. The intersections in Figure 13 used to train the multi-env regime and the intersections in Figure 14 are used in the testing of the multi-env regime.

In order to analyze the performance of the trained model on the same intersection with different set of phases, we consider intersection of Figure 13a in the set of training environments and the intersection of Figure 14a in the test set.

**C.2 Description of Each Environment Instance**

Tables 6 and 7 provides the details of all training and testing problems.

Table 6: Intersections used for the training of the multi-env regime

| case | INT7-H1-4 | INT7-H1-8 | INT1-H1-3 | INT3-H1-2 | INT7-A1-8 | INT7-A2-8 | INT7-A3-8 | INT7-A4-8 | INT7-A5-8 |
|---|---|---|---|---|---|---|---|---|---|
| Intersection | INT7 | INT7 | INT1 | INT3 | INT7 | INT7 | INT7 | INT7 | INT7 |
| #phase | 4 | 8 | 3 | 2 | 8 | 8 | 8 | 8 | 8 |
| traffic | H1 | H1 | H1 | H1 | A1 | A2 | A3 | A4 | A5 |
| #roads | 4 | 4 | 3 | 3 | 4 | 4 | 4 | 4 | 4 |
| #lane | 2 | 2 | 2 | 2 | 2 | 2 | 2 | 2 | 2 |
| case | INT7-A1-4 | INT7-A2-4 | INT7-A3-4 | INT7-A4-4 | INT7-A5-4 | INT8-H1-3 | INT9-S6-4 | INT9-S6-8 | INT1-A1-3 |
| Intersection | INT7 | INT7 | INT7 | INT7 | INT7 | INT8 | INT9 | INT9 | INT1 |
| #phase | 4 | 4 | 4 | 4 | 4 | 3 | 4 | 8 | 3 |
| traffic | A1 | A2 | A3 | A4 | A5 | H1 | S6 | S6 | A1 |
| #roads | 4 | 4 | 4 | 4 | 4 | 4 | 4 | 4 | 3 |
| #lane | 2 | 2 | 2 | 2 | 2 | 2 | 3 | 3 | 2 |
| case | INT1-A2-3 | INT1-A3-3 | INT1-A4-3 | INT1-A5-3 | INT3-A1-2 | INT3-A2-2 | INT3-A3-2 | INT3-A4-2 | INT3-A5-2 |
| Intersection | INT1 | INT1 | INT1 | INT1 | INT3 | INT3 | INT3 | INT3 | INT3 |
| #phase | 3 | 3 | 3 | 3 | 2 | 2 | 2 | 2 | 2 |
| traffic | A2 | A3 | A4 | A5 | A1 | A2 | A3 | A4 | A5 |
| #roads | 3 | 3 | 3 | 3 | 3 | 3 | 3 | 3 | 3 |
| #lane | 2 | 2 | 2 | 2 | 2 | 2 | 2 | 2 | 2 |
| case | INT8-A1-3 | INT8-A2-3 | INT8-A3-3 | INT8-A4-3 | INT8-A5-3 | INT9-S1-4 | INT9-S2-4 | INT9-S3-4 | INT9-S4-4 |
| Intersection | INT8 | INT8 | INT8 | INT8 | INT8 | INT9 | INT9 | INT9 | INT9 |
| #phase | 3 | 3 | 3 | 3 | 3 | 4 | 4 | 4 | 4 |
| traffic | A1 | A2 | A3 | A4 | A5 | S1 | S2 | S3 | S4 |
| #roads | 4 | 4 | 4 | 4 | 4 | 4 | 4 | 4 | 4 |
| #lane | 2 | 2 | 2 | 2 | 2 | 3 | 3 | 3 | 3 |
| case | INT9-S5-4 | INT9-S4-8 | INT9-S5-8 | INT9-S1-8 | INT9-S2-8 | INT9-S3-8 | | | |
| Intersection | INT9 | INT9 | INT9 | INT9 | INT9 | INT9 | | | |
| #phase | 4 | 8 | 8 | 8 | 8 | 8 | | | |
| traffic | S5 | S4 | S5 | S1 | S2 | S3 | | | |
| #roads | 4 | 4 | 4 | 4 | 4 | 4 | | | |
| #lane | 3 | 3 | 3 | 3 | 3 | 3 | | | |

# D Other Application Area for Future Research

## D.1 Assemble-to-Order Systems

This problem refers to the situation where parts of different types need to be assembled into a few known finished products, and the goal is to find the sequence of the products to assemble. Since the number and type of input parts may vary among different products, the problem does not have a fixed-sized input. Moreover, the number of orders at each time changes due to the stochasticity of the customer's demand. So, the number of output products does not have a fixed-size.

AttendLight can be utilized to solve this problem. The state-attention extracts the state of each product, reflecting the availability of required parts, numbers of possible finished products, etc. The action-attention is responsible for deciding what to produce at each time with the goal of minimizing objectives total make-span, tardiness, and delay time.

## D.2 Dynamic Matching Problem

In this problem, entities of different types arrive at the system and wait in the queue until they are matched together. Once a matching happens, the entities immediately leave the system and

(a) INT1

(b) INT3

(c) INT7, 4-phases

(d) INT7, 8-phases

(e) INT9, 4-phases

(f) INT9, 8-phases

(g) INT8

Figure 13: All intersection topologies with their available phases which used for the training of the multi-env regime

(a) INT2

(b) INT4

(c) INT5

(d) INT6

(e) INT10, 4-phases

(f) INT10, 8-phases

(g) INT11, 4-phases

(h) INT11, 5-phases

Figure 14: All intersection topologies with their available phases which used for testing the multi-env regime

Table 7: Intersections used for the testing of the multi-env regime

| case | INT7-H2-4 | INT7-H3-4 | INT7-H4-4 | INT7-H5-4 | INT7-H2-8 | INT7-H3-8 | INT7-H4-8 | INT7-H5-8 | INT2-A4-3 |
|---|---|---|---|---|---|---|---|---|---|
| Intersection | INT7 | INT7 | INT7 | INT7 | INT7 | INT7 | INT7 | INT7 | INT2 |
| #phase | 4 | 4 | 4 | 4 | 8 | 8 | 8 | 8 | 3 |
| traffic | H2 | H3 | H4 | H5 | H2 | H3 | H4 | H5 | A4 |
| #roads | 4 | 4 | 4 | 4 | 4 | 4 | 4 | 4 | 3 |
| #lane | 2 | 2 | 2 | 2 | 2 | 2 | 2 | 2 | 2 |

| case | INT2-A3-3 | INT2-A5-3 | INT2-A1-3 | INT2-H1-3 | INT4-A4-3 | INT4-A3-3 | INT4-A5-3 | INT4-A1-3 | INT4-H1-3 |
|---|---|---|---|---|---|---|---|---|---|
| Intersection | INT2 | INT2 | INT2 | INT2 | INT4 | INT4 | INT4 | INT4 | INT4 |
| #phase | 3 | 3 | 3 | 3 | 3 | 3 | 3 | 3 | 3 |
| traffic | A3 | A5 | A1 | H1 | A4 | A3 | A5 | A1 | H1 |
| #roads | 3 | 3 | 3 | 3 | 3 | 3 | 3 | 3 | 3 |
| #lane | 2 | 2 | 2 | 2 | 3 | 3 | 3 | 3 | 3 |

| case | INT11-S2-3 | INT11-S4-3 | INT11-S3-3 | INT11-S5-3 | INT11-S1-3 | INT11-S6-3 | INT11-S2-5 | INT11-S4-5 | INT11-S3-5 |
|---|---|---|---|---|---|---|---|---|---|
| Intersection | INT11 | INT11 | INT11 | INT11 | INT11 | INT11 | INT11 | INT11 | INT11 |
| #phase | 3 | 3 | 3 | 3 | 3 | 3 | 5 | 5 | 5 |
| traffic | S2 | S4 | S3 | S5 | S1 | S6 | S2 | S4 | S3 |
| #roads | 4 | 4 | 4 | 4 | 4 | 4 | 4 | 4 | 4 |
| #lane | 3 | 3 | 3 | 3 | 3 | 3 | 3 | 3 | 3 |

| case | INT11-S5-5 | INT11-S1-5 | INT11-S6-5 | INT10-S2-4 | INT10-S4-4 | INT10-S3-4 | INT10-S5-4 | INT10-S1-4 | INT10-S6-4 |
|---|---|---|---|---|---|---|---|---|---|
| Intersection | INT11 | INT11 | INT11 | INT10 | INT10 | INT10 | INT10 | INT10 | INT10 |
| #phase | 5 | 5 | 5 | 4 | 4 | 4 | 4 | 4 | 4 |
| traffic | S5 | S1 | S6 | S2 | S4 | S3 | S5 | S1 | S6 |
| #roads | 4 | 4 | 4 | 4 | 4 | 4 | 4 | 4 | 4 |
| #lane | 3 | 3 | 3 | 3 | 3 | 3 | 3 | 3 | 3 |

| case | INT10-S2-8 | INT10-S4-8 | INT10-S3-8 | INT10-S5-8 | INT10-S1-8 | INT10-S6-8 | INT4-S1-3 | INT4-S2-3 | INT4-S3-3 |
|---|---|---|---|---|---|---|---|---|---|
| Intersection | INT10 | INT10 | INT10 | INT10 | INT10 | INT10 | INT4 | INT4 | INT4 |
| #phase | 8 | 8 | 8 | 8 | 8 | 8 | 3 | 3 | 3 |
| traffic | S2 | S4 | S3 | S5 | S1 | S6 | S1 | S2 | S3 |
| #roads | 4 | 4 | 4 | 4 | 4 | 4 | 4 | 4 | 4 |
| #lane | 3 | 3 | 3 | 3 | 3 | 3 | 3 | 3 | 3 |

| case | INT4-S4-3 | INT4-S5-3 | INT4-S6-3 | INT5-A1-3 | INT5-A3-3 | INT5-A4-3 | INT5-A5-3 | INT5-H1-3 | INT5-S1-3 |
|---|---|---|---|---|---|---|---|---|---|
| Intersection | INT4 | INT4 | INT4 | INT5 | INT5 | INT5 | INT5 | INT5 | INT5 |
| #phase | 3 | 3 | 3 | 3 | 3 | 3 | 3 | 3 | 3 |
| traffic | S4 | S5 | S6 | A1 | A3 | A4 | A5 | H1 | S1 |
| #roads | 4 | 4 | 4 | 4 | 4 | 4 | 4 | 4 | 4 |
| #lane | 3 | 3 | 3 | 3 | 3 | 3 | 3 | 3 | 3 |

| case | INT5-S2-3 | INT5-S3-3 | INT5-S4-3 | INT5-S5-3 | INT5-S6-3 | INT6-A1-3 | INT6-A3-3 | INT6-A4-3 | INT6-A5-3 |
|---|---|---|---|---|---|---|---|---|---|
| Intersection | INT5 | INT5 | INT5 | INT5 | INT5 | INT6 | INT6 | INT6 | INT6 |
| #phase | 3 | 3 | 3 | 3 | 3 | 3 | 3 | 3 | 3 |
| traffic | S2 | S3 | S4 | S5 | S6 | A1 | A3 | A4 | A5 |
| #roads | 4 | 4 | 4 | 4 | 4 | 4 | 4 | 4 | 4 |
| #lane | 3 | 3 | 3 | 3 | 3 | 3 | 3 | 3 | 3 |

| case | INT6-H1-3 | INT6-S1-3 | INT6-S2-3 | INT6-S3-3 | INT6-S4-3 | INT6-S5-3 | INT6-S6-3 |
|---|---|---|---|---|---|---|---|
| Intersection | INT6 | INT6 | INT6 | INT6 | INT6 | INT6 | INT6 |
| #phase | 3 | 3 | 3 | 3 | 3 | 3 | 3 |
| traffic | H1 | S1 | S2 | S3 | S4 | S5 | S6 |
| #roads | 4 | 4 | 4 | 4 | 4 | 4 | 4 |
| #lane | 3 | 3 | 3 | 3 | 3 | 3 | 3 |

some feedback is observed in terms of a reward signal. The objective is to maximize the long-term cumulative reward while keeping the system stable. This problem is a generalization of the assemble-to-order systems, where the entities can be humans, advertisements, commodities, etc. The AttendLight model can be used in this problem to first, extract the state of each matching option from the system state, and then the action attention chooses the matchings with the highest value.

### D.3 Wireless Resource Allocation

In this problem, the goal is to efficiently allocate the spectrum and power of the wireless router to its users. Each user may send or receive packets of different sizes. Users and as well as packets may have different priorities. Therefore, the main task is to send the packets considering their priority, size, arrival time, etc. to minimize a relevant cost function such as the queue length or the total response time of each request. AttendLight can solve this problem: first, state-attention learns to extract the state of each packet with considering the arrival time, priority, size of the packet, type of the data, and other information that may not be available for all packets. Then, the action-attention decides to send which packet next.