[Reviews · NeurIPS 2020]

Review 1

Summary and Contributions: This paper addresses the problem of training an RL controller to manage the signals at a traffic intersection with the aim of reducing average travel time. Exisiting approaches to this problem have only considered a single interesection and structure meaning that controllers must be re-trained for each new intersection. This paper introduces an attention-based mechanism for training a single controller that can be applied to various intersection structures without re-training. An empirical study shows that the approach leads to lower average travel time on both training and held-out intersections. Overall, I like the paper. I do feel that it might be too application driven and not general enough for NeurIPS. For that reason I'm leaning towards below acceptance threshold.

Strengths: This work will be of interest to people working at the intersection of RL and traffic management. It may also be of interest to people working on generalization in RL when generalization must be done to problems with different inputs / outputs. The work is novel and presents a well-perfoming solution to a real world application. The experiments are well done and show the method outperforms the considered baselines.

Weaknesses: Limited interest to wider ML community. It may be too application focused for a general ML conference. As pointed out by the authors, transfer learning is a baseline approach for this problem but the empirical comparison did not consider transfer learning. This comparison seems important. There was not much discussion of the training costs (i.e., sample complexity). The paper seems to focus more on "can we learn to generalize intersection control?" than "how to learn generalizable intersection control efficiently?" The latter seems very important for real world application.

Correctness: Yes. The experiments are well done. I skimmed supplemental material and think the author's made a good effort towards reproducibility.

Clarity: Yes it is clearly written.

Relation to Prior Work: Yes, the authors discuss other approaches and how what they do relate.

Reproducibility: Yes

Additional Feedback: If the authors can add discussion of how there approach could be applied to other problems I think it would increase the impact of this paper. I think the paper is borderline as-is but have raised my score as I'm fine with it being accepted. ############### Before Response and discussion ######### Was there a reason not to compare to transfer methods? Can you add discussion on how the sample complexity of the proposed approach and baselines compares? Minor point: could you discuss potential negative impacts in broader impact statement? It seems possible that the control policy might learn something undesirable and it would be interesting to have your thoughts on if / how this could happen. Suggested related work: "Learning an Interpretable Traffic Signal Control Policy" "Traffic signal optimization through discrete and continuous reinforcement learning with robustness analysis in downtown Tehran" "Deep Learning vs. Discrete Reinforcement Learning for Adaptive Traffic Signal Control"


Review 2

Summary and Contributions: This paper proposes a DRL method for single-intersection traffic signals control. The novel aspect of the method is that it is universal model that can be applied to intersections with any number of road, incoming lanes, phases, etc. This is achieved through incorporating two attention models (for state and action resp.) in the policy network, which is trained by REINFORCE. Benchmarking simulation results show that the proposed method suffers small degradation in performance in a multi-env setting w.r.t. single-env, and the proposed method compares favorably to all the baselines.

Strengths: - The use of the two attention models smartly solves the problem of heterogeneity in the number of lanes, phases, etc across different intersection configurations for the resulting method to be universal. - The explanation of the model is very clear. - Empirical results show strong support to the effectiveness of the universal model.

Weaknesses: - Some of the components in the proposed policy network can be better justified through ablation studies. For example, the mean operation used as the 'query' vector in the state attention model, and the LSTM cell used in the action attention model.

Correctness: I believe the methodology is sound.

Clarity: The paper is very clear and well written. I enjoyed reading it.

Relation to Prior Work: This is discussed in details in the related work section.

Reproducibility: Yes

Additional Feedback: Fig 3: It is better to add a note to the stacked bar plot that the color on top of another has a longer ATT than the one below it. It might be different from what one expects for a regular stacked plot. Fig 5: Is there any difference in distribution of \rho_m for training instances v.s. testing instances? Is there any pattern in the instances that yield positive \rho_m's? Have the authors ever analyzed the resulting attention variables to see if it can explain any pattern in the state?


Review 3

Summary and Contributions: This paper introduces an attention-based architecture for traffic signal control. The architecture contains two modules, the first one outputs a representation for a given phase of the intersection (a phase is a set of compatible lanes/trajectories). The second one takes the representation of the various phases and outputs a policy for which phase to select (ie shift the light to green). The policy is trained using Reinforce, and aimed at minimizing a proxy for average travel time for the cars close to the intersection. The architecture is compared to existing algorithms and compares very favorably. Additionally, the attention architecture allows to handle arbitrary interesection topologies out of the box, which is a very useful feature. Experiments show that the trained policy can generalize to unseen intersections.

Strengths: Novel application of attention-based networks to traffic light control, which is an interesting, potentially impactful direction. The universality of the architecture allows out-of-the box application of a trained policy to new intersections, without the need for adaptation. It is a purely empirical paper, the evaluation is convincing. Below are a few suggestions on how to improve it. I am also interested in future applications to network of intersections where green wave effects can be quite important.

Weaknesses: The paper is purely empirical. A few additional experiments could strengthen it: - evaluate CO2 emissions from existing models plugged into the simulator, and demonstrate gains from the AttendLight - bouncing on the observation that performance improves on some test intersections, I would suggest an additional fine-tuning on test intersections (separately), to see if there's some positive transfer. - one interesting ablation / baseline: just average state representations instead of the state-attention module. To see where the gains come from. In terms of broader impact, demand is fairly elastic, so improving signaling is good but not a silver bullet either.

Correctness: Yes

Clarity: Yes, see minor details below.

Relation to Prior Work: It is.

Reproducibility: Yes

Additional Feedback: Post-rebuttal: thanks for the reply and the extra experiments, keeping my score as is. ---------------------------------- Out of curiosity, what makes intersection (d) in Fig.3 so special (MaxPressure is terrible there while FixedTime performs well)? The 5 phases? ----- Minor details Line 13: we show cover intersections Line 40-41: which has flourished Line 90-91: apply transfer learning to Line 93: of transfer learning. Line 136: show Line 152: was introduced Line 153: healthcare [4] (missing space) Line 154: systems [17] (missing space) Line 199: A.3 Line 214: a combination of real-world and ... Line 223: the the Line 239: For the purpose of comparison


Review 4

Summary and Contributions: The authors address the issue of avoiding model adjustment for every intersection through the use of attention. The model is trained once on a variety of intersection configurations. It can then be used to control intersections with the same configuration, but differing traffic patterns than seen in the training set. An increase in average travel time is observed across the new traffic patterns in comparison to retraining for each new pattern.

Strengths: Experiments are extensive and include some real-world scenarios. The method for using attention in the TSCP is clearly explained.

Weaknesses: 1. The primary motivation for the work is not well supported. Certainly, cities do manage thousands of intersections. While unquantified, it is not clear that the cost of training individually would surpass that of the degradation seen in the multi-env setting. 2. It is stated both that the multi-env model has an inevitable performance loss and that the multi-env model outperforms the single-env model due to knowledge sharing. These two statements seem to be conflicting. Please clarify. 3. In section 5.1, the single-env results, it is not clear that FRAP is only applicable in 37 of the 112 cases. As there is quite a lot of recent work on the single-env TSCP. It would have been better to compare to a less restrictive baseline. Such methods can be found in the following: a. Shabestary, Soheil Mohamad Alizadeh, and Baher Abdulhai. "Deep learning vs. discrete reinforcement learning for adaptive traffic signal control." International Conference on Intelligent Transportation Systems (ITSC). IEEE, 2018. b. Ault, James, et al. "Learning an Interpretable Traffic Signal Control Policy." International Conference on Autonomous Agents and MultiAgent Systems. AAMAS, 2020. c. Liang, Xiaoyuan, et al. "Deep reinforcement learning for traffic light control in vehicular networks." IEEE Transactions on Vehicular Technology. IEEE, 2019. 4. In the supplementary material it is stated: “AttendLight in single-env regime outperforms other algorithms in 107 cases out of 112 cases“ and that AttendLight reduces ATT by 10% on average over FRAP. While it is clear how attention is useful in the multi-env setting, could you please add some analysis as to why it is expected to outperform an algorithm designed for single intersections? 5. As it is proposed in the paper that the method is suitable for city-wide control, it is important to provide an analysis of the worst-case results of the method. If on average traffic is alleviated, but certain intersections become nearly impassable this would not be a viable solution. A glance at the numbers in the supplement shows this method may result in some intersections experiencing a 78% increase in average travel time. Please provide such a worse case analysis.

Correctness: Correct

Clarity: Clear

Relation to Prior Work: Should be improved. See "Weaknesses"

Reproducibility: Yes

Additional Feedback: Minor Line 228: typo - moving vehicles in “chunk” Questions for authors Why should the degradation of multi-env be worse on the training-set (case 1) than the testing-set (case 2)? After rebuttal: Change (i) the added tuning process is not novel as it is similar to the one in MetaLight[28]. Also, doesn't this addition removes the supposed advantage of AttendLight? The authors did not address my concerns regarding worst case performance. I still believe that this is a borderline paper for NeurIPS.

[Author Response · NeurIPS 2020]

Thank you for carefully reviewing the manuscript and finding our idea *"novel"* and *"interesting"*.

**Major Changes**: (i) As suggested by R#3, we have added a new method in multi-env regime, which fine-tunes the policy on the fly. In this method, we start with the multi-env policy and improve with a very few training steps to calibrate the policy for every specific intersection. Following that, the maximum and average of multi-env gap compared to single-env decreased significantly after 200 training episodes (instead of 100,000 episodes when trained from scratch) such that we got to 5% gap in average with respect to the single-env regime. After 1000 training steps this gap decreased to 3%. (See the first row of the figures). (ii) We added a new benchmark, DQTSC-M, (R#1 and R#4). As it is shown in the second row of the figure, single-env model obtains 15% smaller ATT in average and outperforms DQTSC-M in all 112 cases. Compared to the multi-env regime (the third row of the figures), AttendLight obtains 4% smaller ATT compared to DQTSC-M. (iii) Added the result of AttendLight with mean-state instead of state-attention model (R#2 and R#3).

**R#1**: Thank you for carefully reviewing the paper and finding it "novel".

- *"too application focused"*: There is a growing interest in application-focused papers in NeurIPS as ML/RL is becoming common in real-world. There is an "application" subject area where $\sim 20\%$ of accepted papers fall under this category [1] in the NeurIPS 2019. Besides, relevant papers on traffic control are published in NeurIPS. For example, see [2]. Moreover, AttendLight framework by itself is of independent interest and can be applied to various domains such as matching, routing, etc. We will add a discussion about this.

- *"transfer learning baselines"*: In the current literature of TSCP, transfer learning is used for reducing the training time and it usually results in the same or even worse performance. For example, FRAP uses transfer learning to train policies for new intersections which achieves the same quality solution in comparison to the training from scratch (see, Figure 10 in [30]). Instead, we have compared AttendLight with the existing fully trained from scratch RL-based alternatives and we don't expect that adding transfer learning will affect the solution quality of the baselines. We will verify the validity of this claim in our camera-ready version.

- *"training costs"*: Some training details are explained in appendix A.3. We will add more details in the final version.

- *"learn generalizable intersection control efficiently"*: We added a fine-tuning mechanism to further improve the generalizability of the multi-env results. Following this makes the multi-env regime quite efficient in terms of training-time. See the Major change comment.

**R#2**: Thank you for your positive feedback and great suggestions.

- *"ablation studies"*: We added one ablation study to show the effect of state-attention (as suggested by R#3). The idea behind having mean query was to learn the importance of each lane-traffic compared to the average traffic per lane. We will clarify these points in final revision and add experiments to better justify the role of components.

- *"distribution of $\rho_m$ "*: As it can be seen from Figure 4, there is no noticeable difference between these two groups. We will add the separated figures of $\rho_m$ for training and testing to the appendix.

- *"pattern in the state"*: This is a great suggestion. We will try to add a section about this to the final version.

**R#3**: Thank you for carefully reviewing the paper and your great suggestions.

- *"CO2 emissions"*: CityFlow does not provide CO2 statistics, but we will construct a CO2 emission metric based on the traffic flow, and will add the reduction of CO2 emission to the appendix.

- *"fine-tuning"*: This is a great suggestion. After fine-tuning for 200 episodes, the average gap drops to 3% (from 13%) and the worst-case gap is decreased to 21%. See the major changes.

- *"ablation Study"*: We re-ran the single-env regime for all cases with the mean-state. The results show that mean-state obtains 5% larger ATT (in average) than that of with state-attention. We will add this result to the appendix.

**R#4**: We appreciate your constructive feedback and finding the paper "novel".

- *"primary motivation"*: To reduce the degradation observed in mutli-env regime, we added a fine-tuning which helps the multi-env regime to fine-tune the policy quickly. See major changes.

- *"conflicting statements..."*: In average, multi-env performs worse than single-env; although, there are some special cases that the multi-env model outperforms the single-env model. We will make sure to remove the confusion.

- *"less restrictive baseline"*: Thanks for introducing these papers. We added DQTSC-M. See the major change above.

- *"why single-env AttendLight outperforms FRAP"*: First, we would like to mention that AttendLight supports both single-env and multi-env by design. We believe that the superior performance of single-env model compared to FRAP originates from two attention model.

- *"city-wide control performance..."* After using the quick fine-tuning, the ATT gap of the intersection that you mentioned is now 5%, after 200 training-episodes. Further training to 1000 episodes decreases the gap to 3%.

[1] What we learned from neurips 2019 data. https://medium.com/@NeurIPSConf/what-we-learned-from-neurips-2019-data-111ab996462c. Accessed: 2020-08-12.

[2] Silvia Richter, Douglas Aberdeen, and Jin Yu. Natural actor-critic for road traffic optimisation. In *Advances in neural information processing systems*, pages 1169–1176, 2007.


[Meta-Review · NeurIPS 2020]

There was a consensus among reviewers that the paper should be accepted. The paper provides a novel application of attention-based networks to traffic light control. The universality of the architecture allows out-of-the box application of a trained policy to new intersections without the need for adaptation. Overall, this paper seems to have the potential to have a strong impact.